# Reconstruction of Cortical Surfaces with Spherical Topology from Infant Brain MRI via Recurrent Deformation Learning

## Abstract

Cortical surface reconstruction (CSR) from MRI is key to investigating brain structure and function. While recent deep learning approaches have significantly improved the speed of CSR, a substantial amount of runtime is still needed to map the cortex to a topologically-correct spherical manifold to facilitate downstream geometric analyses. Moreover, this mapping is possible only if the topology of the surface mesh is homotopic to a sphere. Here, we present a method for simultaneous CSR and spherical mapping efficiently within seconds. Our approach seamlessly connects two sub-networks for white and pial surface generation. Residual diffeomorphic deformations are learned iteratively to gradually warp a spherical template mesh to the white and pial surfaces while preserving mesh topology and uniformity. The one-to-one vertex correspondence between the template sphere and the cortical surfaces allows easy and direct mapping of geometric features like convexity and curvature to the sphere for visualization and downstream processing. We demonstrate the efficacy of our approach on infant brain MRI, which poses significant challenges to CSR due to tissue contrast changes associated with rapid brain development during the first postnatal year. Performance evaluation based on a dataset of infants from 0 to 12 months demonstrates that our method substantially enhances mesh regularity and reduces geometric errors, outperforming state-of-the-art deep learning approaches, all while maintaining high computational efficiency.

## 1 Introduction

The cerebral cortex is a prominent part of the human brain with distinct regions responsible for functions such as perception, language, and cognition. Analyzing the cortical surface can be challenging since it is highly folded, characterizing gyri with ridges and sulci with valleys (Fischl, 2012).

Software packages developed for cortical analysis, such as BrainSuite (Shattuck & Leahy, 2002), FreeSurfer (Fischl, 2012), Connectome Workbench (Glasser et al., 2013), Infant FreeSurfer (Zöllei et al., 2020), and iBEAT V2.0 (Wang et al., 2023), typically involve multiple steps (Fig. 1): inhomogeneity correction, skull stripping, tissue segmentation, hemisphere separation, subcortical filling, topology correction, surface reconstruction, feature computation, spherical mapping, registration, and parcellation. Certain steps can be time consuming, making application of these tools to large datasets challenging. For example, FreeSurfer (Dale et al., 1999) takes several hours to reconstruct the cortical surfaces and about half an hour for surface inflation and spherical mapping from a single MRI scan.

CSR can be expedited with implicit surface representation methods (Gopinath et al., 2021; Cruz et al., 2021), multi-layer perceptrons (Ma et al., 2021), and graph convolutional neural networks (Hoopes et al., 2022; Bongratz et al., 2022). However, these approaches lack guarantees of anatomically valid surface predictions, leading to surfaces with topological defects. This poses challenges for downstream tasks, as a mesh must have a homotopic topology to that of a sphere for successful diffeomorphic inflation and spherical mapping (Hoopes et al., 2022).

Flow-based CSR methods (Lebrat et al., 2021; Santa Cruz et al., 2022; Ma et al., 2022; Zheng et al., 2023; Chen et al., 2023; Ma et al., 2023) have recently gained prominence primarily due to their ability to ensure topology correctness and their remarkable time efficiency, significantly reducing CSR time from several hours to just a few seconds. While these methods vary in architectural design,

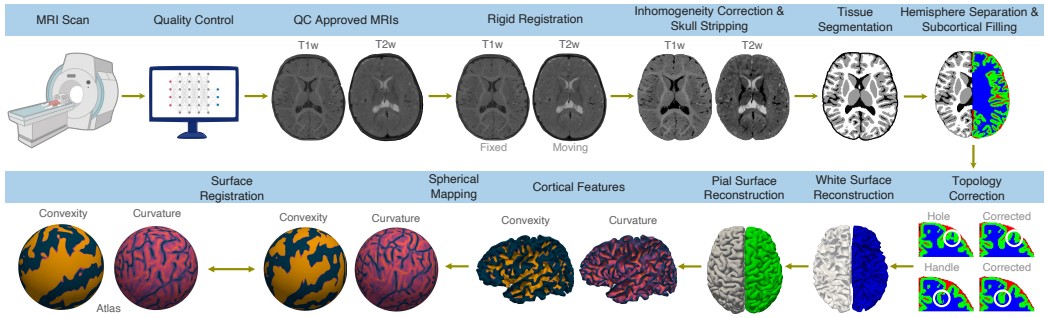

Figure 1: Typical pipeline for cortical surface analysis.

initial mesh, loss functions, and target cohorts, they typically employ a flow ordinary differential equation (ODE) to model mesh deformation, assuming a stationary vector field (SVF) inferred from an image using a deep neural network. CortexODE (Ma et al., 2022) and SurfNN (Zheng et al., 2023) deform an initial surface reconstructed from a tissue segmentation map after subcortical filling and topology correction. CorticalFlow (Lebrat et al., 2021), CorticalFlow++ (Santa Cruz et al., 2022), SurfFlow (Chen et al., 2023), and CoTAN (Ma et al., 2023) deform either a convex hull or an inflated white surface. However, they require an additional time-consuming spherical mapping step to transform surfaces to the geometry of a sphere. This transformation enables curvature-matching registration with an average atlas (Fischl, 2012) and, consequently, establishes a global anatomical mapping across surfaces, facilitating group-level analysis and surface segmentation (Hoopes et al., 2022).

Here, we present an efficient flow-based approach that integrates CSR and spherical mapping in a unified framework. Our method involves deforming a spherical template mesh, encompassing the largest brain in our dataset, to match a target cortical surface. This ensures one-to-one vertex correspondence between a cortical surface and a sphere, enabling direct spherical mapping without additional time costs. We introduce a recurrent strategy to learn large deformations from the sphere to the intricately folded white and pial surfaces while preserving mesh topology. To maintain mesh uniformity and account for changing brain volumes, we propose a novel adaptive edge length loss, preventing unwanted distortions. We validate the effectiveness of our method on the challenging task of CSR from the brain MRI of infants from 0 to 12 months of age. Our method completes CSR and spherical mapping within seconds and substantially improves mesh regularity and reduces geometric errors compared with state-of-the-art deep learning approaches.

## 2 RELATED WORK

### 2.1 TRADITIONAL CSR METHODS

Traditional CSR methods (Shattuck & Leahy, 2002; Fischl, 2012; Glasser et al., 2013; Zöllei et al., 2020; Wang et al., 2023) involve multiple steps (Fig. 1), which can be potentially time consuming, limiting their applicability to large studies.

### 2.2 DEEP LEARNING CSR METHODS

Deep neural networks have been shown to substantially speed up some steps in traditional methods. FastSurfer (Henschel et al., 2020) accelerates FreeSurfer by employing deep learning tissue segmentation and fast spherical mapping, but does not speed up CSR. SegRecon (Gopinath et al., 2021) and DeepCSR (Cruz et al., 2021) predict a signed distance map for implicit surface representation. PialNN (Ma et al., 2021) deforms an initial white surface to the pial surface by predicting vertex-wise displacement vectors using a multi-layer perceptron. TopoFit (Hoopes et al., 2022) and Vox2Cortex (Bongratz et al., 2022) deform a template mesh iteratively using a combination of convolutional and graph convolutional neural networks. While these methods have improved efficiency, they do not provide a guarantee of topological correctness.

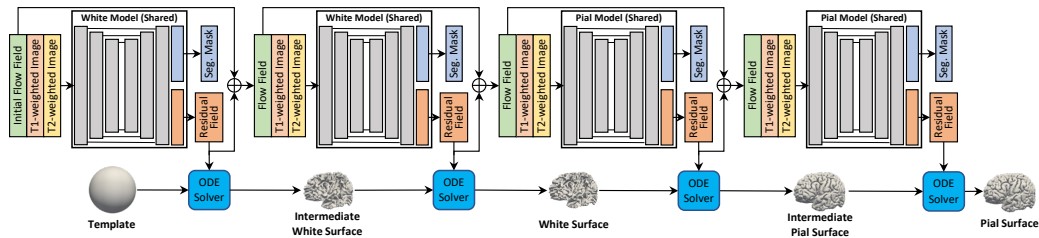

Figure 2: Method overview. The white and pial models iteratively deform a template mesh. They are unfolded for ease of illustration.

Recently, flow-based methods have gained popularity due to their impressive time efficiency and theoretical guarantees in preserving surface topology. Although varying in architecture (cascaded (Lebrat et al., 2021; Santa Cruz et al., 2022; Chen et al., 2023; Ma et al., 2023) or multi-branch (Zheng et al., 2023)), initial mesh (convex hull (Lebrat et al., 2021; Santa Cruz et al., 2022; Chen et al., 2023), a segmentation-based surface (Ma et al., 2022; Zheng et al., 2023), or an inflated white surface (Ma et al., 2023)), loss functions, and target cohorts (neonates (Ma et al., 2023), infants (Chen et al., 2023), or adults (Lebrat et al., 2021; Santa Cruz et al., 2022; Ma et al., 2022; Zheng et al., 2023)), these methods commonly solve an ODE that models the trajectory of each vertex of a surface. Note that the choice of the initial mesh has significant impact on CSR (e.g., topological correctness) and downstream applications (e.g., necessity of spherical mapping).

Our method employs multiple SVFs for diffeomorphic deformation of a spherical template mesh. Unlike Voxel2Mesh (Wickramasinghe et al., 2020) and MeshDeformNet (Kong et al., 2021), which caters to organs with simpler shapes (e.g., the hippocampus, liver, and heart), our task is substantially more challenging with the need to deform a spherical template mesh to the highly folded cortex. We introduce a recurrent strategy with multiple SVFs for greater representation capacity to learn the large deformation from the template mesh to the white and pial surfaces. We leverage the one-to-one vertex correspondence between the template mesh and the cortical surfaces to facilitate straightforward spherical mapping.

## 3 METHODS

### 3.1 ARCHITECTURE

Our model (Fig. 2) comprises two sub-networks with a common U-shaped architecture: a white model and a pial model. Both models operate recurrently, predicting a SVF each time to estimate per-vertex displacement vectors building on the current state of deformation and the probability maps for brain tissue segmentation. The detail of underlying U-shaped architecture is shown in Fig. S1 in supplementary material. These two networks are seamlessly integrated, allowing the initial spherical template mesh to deform first to the white surface and subsequently to to the pial surface. To manage memory usage, we limit the number of iterations to 2 for each model.

### 3.2 RECURRENT DEFORMATION LEARNING

Deforming a spherical template mesh to the cortical surfaces involves large coarse-to-fine deformations. To overcome the limited capacity of a SVF in handling large deformations, we introduce a recurrent deformation learning strategy to allow large deformations to be learned incrementally with multiple smaller deformations. Each deformation is obtained by ODE integration based on a SVF over time interval $[0, 1]$.

In the $i$-th iteration, model $\mathbf{M}_i$ takes as input a pair of T1-weighted (T1w) image, $\mathcal{I}_{T1}$ and T2-weighted (T2w) image, $\mathcal{I}_{T2}$, and a cumulative velocity field, $\sum_{j=0}^{i-1} \mathbf{V}_j$, obtained by summing voxel-wise velocity fields from previous iterations. SVF $\mathbf{V}_j$, predicted by $\mathbf{M}_j$, is a dense map with each voxel containing a 3D vector. An initial null flow field $\mathbf{V}_0 = \mathbf{0}$ is fed to $\mathbf{M}_1$.

The governing equation for the diffeomorphic deformation in each iteration is an ODE:

$$\frac{\partial \phi_i(t; \mathbf{x}_{ik})}{\partial t} = \mathbf{V}_i(\phi_i(t; \mathbf{x}_{ik})), \tag{1}$$

where $\phi_i$, $\mathbf{V}_i$, and $\mathbf{x}_{ik}$ denote the diffeomorphic deformation, the SVF, and the $k$-th mesh vertex of the $i$-th surface $\mathcal{S}_i$, respectively. Formulating a deformation as an ODE ensures that the transformation between two time points is well-behaved and differentiable. Solving the ODE with the initial condition $\phi_i(0; \mathbf{x}_{ik}) = \mathbf{x}_{i-1,k}$ yields per-vertex displacement vectors. These vectors are used to deform the surface $\mathcal{S}_{i-1}$ given by $\mathbf{M}_{i-1}$ to a new surface $\mathcal{S}_i$. Note that the initial surface $\mathcal{S}_0$ is the spherical template mesh. We numerically solve the ODE using the fourth-order Runge-Kutta method with 30 time steps. Formally,

$$\mathbf{V}_i = \mathbf{M}_i \left( \mathcal{I}_{\text{T1}}, \mathcal{I}_{\text{T2}}, \sum_{j=0}^{i-1} \mathbf{V}_j \right) \quad \text{and} \quad \mathcal{S}_i = \text{ODESolver}(\mathcal{S}_{i-1}, \mathbf{V}_i). \tag{2}$$

### 3.3 DUAL-MODAL INPUT

Infants undergo rapid and dynamic changes in brain structure and sizes. Due to the myelination process and variations in tissue composition and water content, infant MRI scans exhibit three distinct phases during the first year of life. In the *infantile phase* ($\leq 5$ months), gray matter (GM) shows higher signal intensity than white matter (WM) in T1w images. The *isointense phase* (5–8 months) witnesses an intensification of WM signal due to myelination, leading to a reduction in contrast between GM and WM in both T1w images and T2w images. In the *early adult-like phase* ($\geq 8$ months), the GM intensity becomes lower than WM in T1w images, similar to the tissue contrast seen in adult MRI scans. These shifts in contrast and the rapid alterations in brain volume and shape within the first postnatal year present significant challenges for CSR. To address these challenges, we propose leveraging the complementary information from both T1w and T2w images to enhance the reliability and performance.

### 3.4 MULTI-TASK LEARNING

Each component model consists of two branches: one for predicting flow field and the other for segmentation mask. By employing brain tissue segmentation as an auxiliary task, the white and pial models are anatomy-aware, capturing cortical surface boundaries better and avoiding predicting surfaces with artifacts.

### 3.5 SPHERICAL MAPPING

Cortical surface-based analysis provides several advantages over volumetric analysis when studying the intricate cerebral cortex. These benefits encompass improved visualization of folded cortical regions, enhanced spatial normalization of the cortex, and precise measurement of cortical properties. A pivotal component of this analysis is spherical mapping, which involves mapping a highly convoluted cortical surface onto a sphere. This process simplifies subsequent procedures, including registration and parcellation, ultimately facilitating more accurate inter-subject comparisons and supporting longitudinal analysis.

In traditional pipelines, spherical mapping is accomplished iteratively to minimize metric distortion, angle distortion, and area distortion. On average, this process takes about 30 minutes, which can pose a significant obstacle to downstream analysis when dealing with large-scale neuroimaging data. In contrast, spherical mapping in our framework is remarkably straightforward. This simplicity is enabled by the shared topology between the spherical template mesh and the cortical surfaces, along with the one-to-one vertex correspondence that exists between them. These inherent properties empower us to execute spherical mapping with ease and reliability.

### 3.6 LOSS FUNCTION

The loss function used in our approach is a weighted combination of Chamfer distance (CD) $L_{\text{cd}}$, an adaptive edge length loss $L_{\text{edge}}$, and a segmentation loss $L_{\text{seg}}$, balanced by two parameters $\lambda_1$ and $\lambda_2$:

$$L_{\text{total}} = L_{\text{cd}} + \lambda_1 L_{\text{edge}} + \lambda_2 L_{\text{seg}}, \tag{3}$$

**Chamfer Loss**    The Chamfer distance (or Chamfer loss) is a metric used to assess the alignment between two point clouds. It calculates the bidirectional distance from each vertex in the predicted mesh, denoted as $P$, to the closest vertex in the ground truth mesh, denoted as $Q$:

$$L_{\text{cd}} = \frac{1}{|P|} \sum_{p \in P} \min_{q \in Q} \|p - q\|_2^2 + \frac{1}{|Q|} \sum_{q \in Q} \min_{p \in P} \|p - q\|_2^2. \tag{4}$$

**Adaptive Edge Length Loss**    To accommodate dynamic changes in brain volume during the first postnatal year and ensure satisfactory mesh regularity, we adaptively constrain the size of triangles in the predicted meshes for each subject based on their brain volume during training. To achieve this, we assume an ideal prediction where the faces are equilateral and of the same size. We calculate the target face area by dividing the total surface area $S$ of the ground truth surface by the number of faces $N$ in the spherical template mesh. Subsequently, the target edge length $\mu_{\text{adaptive}}$ is adaptively set to $2\sqrt{\frac{S}{\sqrt{3}N}}$. We propose using the adaptive edge length loss below to drive the edge length to $\mu_{\text{adaptive}}$.

$$L_{\text{edge}} = \frac{1}{|P|} \sum_{p \in P} \frac{1}{|\mathcal{N}(p)|} \sum_{k \in \mathcal{N}(p)} (\mu_{\text{adaptive}} - \|p - k\|_2)^2 \tag{5}$$

where $P$ denotes the predicted mesh, $p$ is a vertex on $P$, $\mathcal{N}(p)$ are the neighbors of $p$.

**Segmentation Loss**    The segmentation loss is a combination of cross-entropy and the dice loss.

### 3.7    IMPLEMENTATION DETAILS

The network was sequentially trained. We froze the parameters of the white model after training it for 81k iterations and then started training the pial model. The pial model was trained for 54k iterations. Adam optimizer was used and the initial learning rate was set to 0.0001. The loss weights $\lambda_1$ and $\lambda_2$ were set to 1.0 and 1e-3, respectively, to balance the mesh regularity and the quantitative performance. Instance normalization (IN) (Ulyanov et al., 2016) helps reducing co-variate shift in deep networks by normalizing features to zero mean and unit variance across each channel for each observation independently. We observed that the inclusion of IN layers between convolutional and activation layers significantly accelerates convergence.

## 4    RESULTS

### 4.1    DATA

The T1w and T2w images are sourced from the Baby Connectome Project (BCP) (Howell et al., 2019). It comprises 121 subjects ranging in age from 2 weeks to 12 months. The dataset was partitioned into 3 subsets, with 90 cases allocated for training, 12 cases for validation, and 19 cases for testing. Rigid registration was applied to the T1w and T2w image pairs. Pseudo-ground truth cortical surface meshes were generated using iBEAT v2.0 (Wang et al., 2023) for both training and performance evaluation. iBEAT employed deep neural networks for brain tissue segmentation, while adhering to traditional strategies for image registration to an atlas, topological correction, and surface reconstruction using the marching cube algorithm. It's worth noting that the use of pseudo-ground truth as a reference is a common practice in cortical analysis, as adopted by existing CSR methods. The spherical template mesh, which contains $\sim$164k vertices, was generated by iteratively subdividing the faces of an icosahedron.

### 4.2    COMPARISON WITH STATE-OF-THE-ART METHODS

We compared the proposed method with several state-of-the-art baseline methods, including DeepCSR, CorticalFlow++, SurfFlow, Vox2Cortex, and CortexODE, which are considered representative in the field. Chamfer distance (CD), average symmetric surface distance (ASSD), 90% Hausdorff distance (HD), normal consistency (NC) and percentage of self-intersection (SI) were utilized as evaluation metrics.

Table 1: Comparison of different CSR methods in reconstructing the white and pial surfaces of the left (L) and right (R) hemispheres. ↓ and ↑ indicate better performance with lower and higher metric values, respectively.

| | Pial Surface | | | | | | | | | |
|---|---|---|---|---|---|---|---|---|---|---|
| | CD ↓ | | ASSD ↓ | | HD ↓ | | NC ↑ | | SI % ↓ | |
| | L | R | L | R | L | R | L | R | L | R |
| Ours | 0.35 (±0.09) | 0.38 (±0.14) | 0.47 (±0.03) | 0.49 (±0.05) | 0.82 (±0.06) | 0.86 (±0.09) | 0.92 (±0.01) | 0.91 (±0.01) | 0.30 (±0.14) | 0.38 (±0.15) |
| DeepCSR | 4.69 (±2.10) | 3.85 (±2.00) | 1.26 (±0.25) | 1.13 (±0.27) | 5.10 (±1.13) | 4.58 (±1.37) | 0.75 (±0.02) | 0.76 (±0.02) | 0.001 (±0.0005) | 0.002 (±0.0008) |
| CorticalFlow++ | 1.02 (±0.17) | 0.67 (±0.10) | 0.74 (±0.03) | 0.63 (±0.04) | 2.15 (±0.25) | 1.55 (±0.12) | 0.81 (±0.01) | 0.84 (±0.01) | 1.30 (±0.42) | 1.21 (±0.34) |
| SurfFlow | 0.39 (±0.14) | 0.36 (±0.11) | 0.48 (±0.03) | 0.46 (±0.05) | 0.98 (±0.08) | 0.80 (±0.10) | 0.90 (±0.01) | 0.91 (±0.01) | 1.43 (±0.52) | 2.63 (±0.57) |
| Vox2Cortex | 1.56 (±2.42) | 1.94 (±3.14) | 0.82 (±0.41) | 0.88 (±0.46) | 2.04 (±1.12) | 2.56 (±1.58) | 0.82 (±0.07) | 0.81 (±0.06) | 8.15 (±2.22) | 6.47 (±1.60) |
| CortexODE | 1.41 (±0.10) | 1.31 (±0.19) | 0.96 (±0.03) | 0.89 (±0.05) | 2.03 (±0.10) | 2.06 (±0.18) | 0.78 (±0.01) | 0.78 (±0.01) | 0.23 (±0.11) | 0.21 (±0.11) |
| | White Surface | | | | | | | | | |
| | CD ↓ | | ASSD ↓ | | HD ↓ | | NC ↑ | | SI % ↓ | |
| | L | R | L | R | L | R | L | R | L | R |
| Ours | 0.36 (±0.14) | 0.38 (±0.15) | 0.47 (±0.05) | 0.49 (±0.06) | 0.81 (±0.09) | 0.87 (±0.11) | 0.94 (±0.01) | 0.94 (±0.01) | 0.06 (±0.06) | 0.05 (±0.03) |
| DeepCSR | 0.74 (±0.50) | 0.50 (±0.90) | 0.58 (±0.11) | 0.60 (±0.16) | 1.28 (±0.58) | 1.27 (±0.68) | 0.90 (±0.02) | 0.90 (±0.02) | 0.001 (±0.0005) | 0.001 (±0.0006) |
| CorticalFlow++ | 1.13 (±0.43) | 0.91 (±0.12) | 0.86 (±0.04) | 0.78 (±0.04) | 1.96 (±0.11) | 1.66 (±0.13) | 0.72 (±0.13) | 0.73 (±0.04) | 0.11 (±0.06) | 0.12 (±0.07) |
| SurfFlow | 0.37 (±0.12) | 0.41 (±0.15) | 0.47 (±0.05) | 0.49 (±0.06) | 0.86 (±0.10) | 0.90 (±0.12) | 0.93 (±0.01) | 0.92 (±0.01) | 0.38 (±0.13) | 1.43 (±0.43) |
| Vox2Cortex | 1.24 (±1.57) | 1.16 (±1.11) | 0.76 (±0.30) | 0.76 (±0.28) | 1.75 (±0.92) | 1.73 (±0.87) | 0.84 (±0.10) | 0.84 (±0.11) | 11.24 (±5.70) | 10.83 (±5.58) |
| CortexODE | 0.39 (±0.13) | 0.43 (±0.15) | 0.49 (±0.05) | 0.51 (±0.06) | 0.86 (±0.12) | 0.91 (±0.13) | 0.93 (±0.01) | 0.93 (±0.01) | 0.04 (±0.04) | 0.003 (±0.003) |

All baselines are reproduced based on the recommended settings specified in their official implementation and paper. To ensure fair comparison, we modified all baseline methods to take dual-modal inputs. For DeepCSR, we employed the finest possible configuration, generating a $512^3$ 3D grid to predict the signed distance field for surface reconstruction using the marching cubes algorithm. For Vox2Cortex, we trained the network with template containing (~41k vertices). During inference, we used the higher resolution template (~168k vertices) for each structure suggested by the author. For CortexODE, we first trained a model for the ribbon segmentation task (subcortical-filled white matter), then utilized the topologically corrected segmentation results to compute SDF, and finally generated the initial subject-specific mesh for mesh deformation using the marching cube algorithm. We followed the same configuration stated in the paper and used 3 levels of cube sampling. For SurfFlow, we used a smoothed convex hull (~360k vertices) computed from our training set. And for CorticalFlow++, we followed the default settings; retrained using three coarse-to-fine smoothed convex hull templates, with ~30k, ~125k, and ~360k vertices respectively.

Experimental results in Table 1 indicate that our method yields an average CD of 0.35 mm for the left hemisphere and 0.38 mm for the right hemisphere of the pial surface. For the white surface, our method yields a CD of 0.36 mm for the left hemisphere and 0.38 mm for the right hemisphere. In contrast, DeepCSR exhibits an average CD of 4.69 mm for the left hemisphere and 3.85 mm for the right hemisphere of the pial surface. For the white surface, DeepCSR yields an average CD of 0.74 mm for the left hemisphere and 0.50 mm for the right hemisphere. The results indicate a substantial improvement in CD, ranging from 24% to 92%, when comparing our method with DeepCSR. CorticalFlow++ and Vox2Cortex show clear improvements in CD for the pial surface compared with DeepCSR, but worse CD for the white surface. Nonetheless, our method outperforms

both DeepCSR and CorticalFlow++ by a significant margin on both surfaces. CortexODE, while only performing slightly worse than our method in white surface reconstruction, exhibits much larger errors, ranging from $244\%$ to $303\%$, in pial surface reconstruction. SurfFlow, representing a strong baseline, also does not perform as well as our method, especially in white surface reconstruction.

We also computed the Euler characteristic of our method's predicted surfaces and all of them are equal to 2, which means they have the correct genus-zero topology. In Table 1, our method shows a relatively low self-intersection (SI) rate ($\sim 0.3\%$ on pial surfaces and $\sim 0.06\%$ on white surfaces). DeepCSR is expected to exhibit very low SI, as it applies the marching cubes algorithm on a predicted SDF map without guarantee for topology correctness. CortexODE achieves a low SI on the white surface, by employing a process similar to DeepCSR (applying marching cubes on predicted SDF map). However, the SI rate goes up when CortexODE deforms from the white surface to the pial surface. Although the SI of CortexODE is slightly lower than our method, the accuracy of the reconstructed pial surface is subpar.

Consistent with CD, our method shows similar improvements when evaluated using metrics such as ASSD, HD, and NC metrics. Notably, our method outperforms all baseline methods on all three of these metrics.

The subpar performance of DeepCSR can be attributed to the presence of numerous mesh topological artifacts in its predictions. These artifacts are likely a consequence of its use of implicit surface representation, which does not guarantee generating a surface with genus-zero manifold. While CorticalFlow++ performs better than DeepCSR, it tends to cluster vertices in specific local regions, resulting in meshes with poor uniformity. This uneven distribution of vertices may account for its modest performance on our dataset. Vox2Cortex and CortexODE, on the other hand, are capable of generating satisfactory surfaces in general. However, their significantly poorer performance on a few challenging cases has a noticeable impact on their overall results. SurfFlow is the only method that comes close to matching the performance of our approach. Nevertheless, its limitations in generating meshes with uniformly distributed vertices and accurately capturing cortical surface boundaries still result in somewhat inferior performance.

Fig. 3 shows a representative case for visual comparison. Fig. 4 presents error maps from six subjects at various ages. These error maps, computed by measuring the distance of each vertex on the predicted surface to its nearest counterpart on the ground truth surface, provide insights into geometric errors. Due to the difference between number of vertices in GT and predicted surface, error maps were computed in one direction (from GT to predictions). DeepCSR, CorticalFlow, and SurfFlow are expected to have better visual performance than the numerical analysis suggests, since the meshes predicted by these methods have a larger number of vertices.

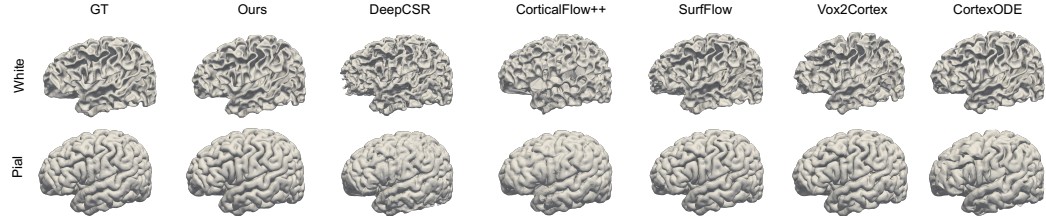

Figure 3: Visual comparison of cortical surfaces reconstructed by various methods.

### 4.3 SPHERICAL MAPPING

Our approach distinguishes itself from existing methods by allowing straightforward spherical mapping by leveraging the inherent one-to-one vertex correspondence between the spherical template mesh and the reconstructed cortical surfaces. To exemplify this advantage, we showcase in Fig. 5 curvature and convexity maps on both the predicted cortical surfaces and the sphere for various time points.

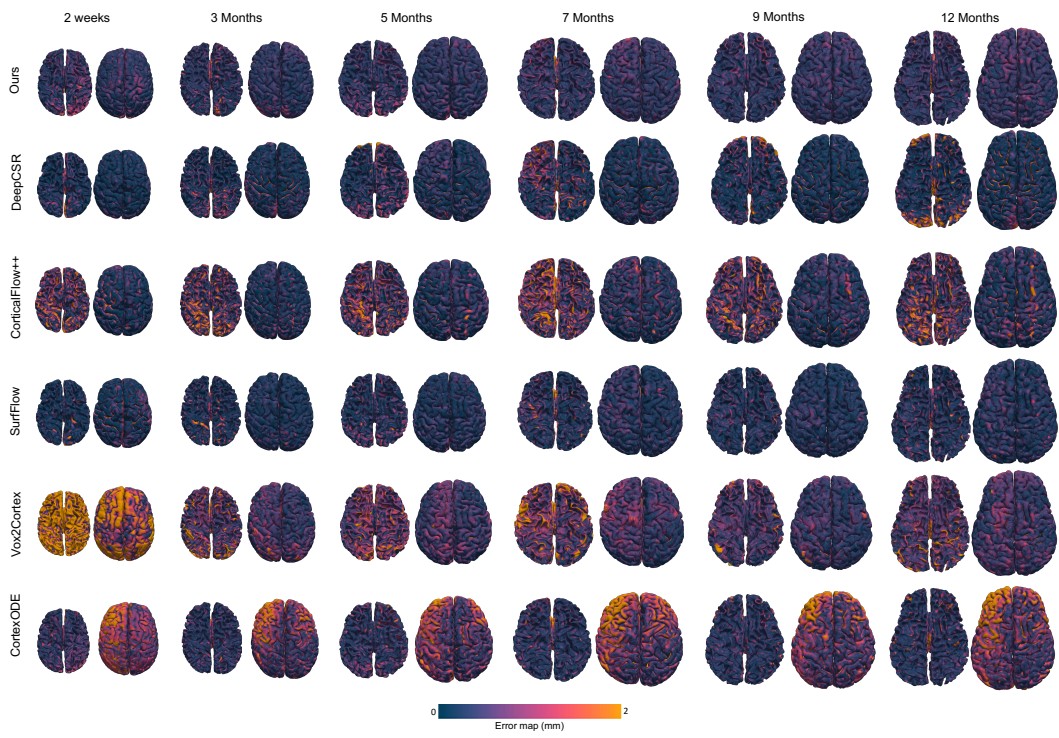

Figure 4: Error maps for white and pial surfaces reconstructed by different methods across age.

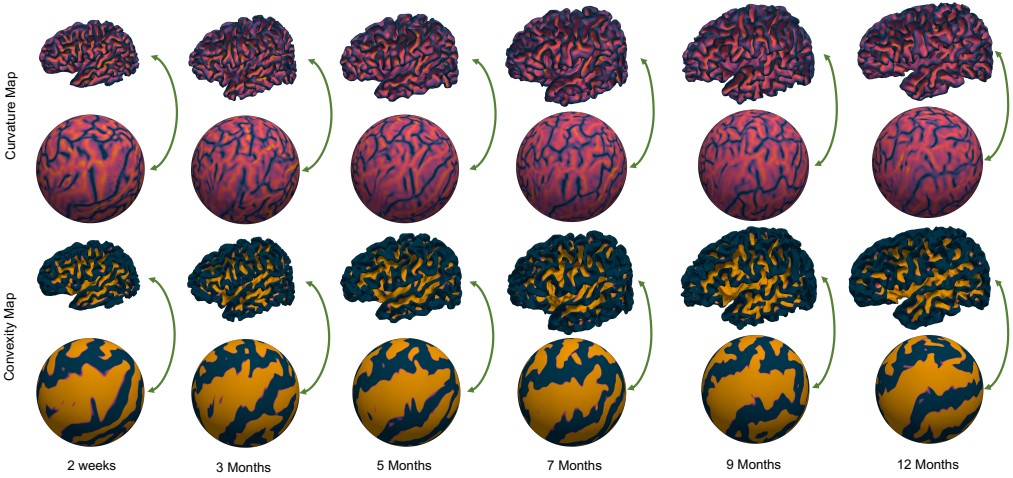

Figure 5: Spherical mapping of curvature and convexity computed with predicted white surfaces.

## 4.4 ABLATION STUDY

We conducted an ablation study to assess the effects of using dual-modal inputs, recurrent learning and the segmentation branch on CSR performance. Quantitative results for different settings are presented in Table 2. It is evident that our dual-modal training substantially reduces errors in CSR from infant brain MRI. The incorporation of the recurrent deformation learning strategy yields noticeable additional improvements, underscoring its utility in deformation learning. The segmentation module, while contributing only slightly to quantitative results, has a more substantial impact on the visual

Table 2: Cortical surface reconstruction performance with respect to modalities (T1w and T2w), recurrent learning (RL) and segmentation module (Seg). Left (L) and right (R) sides of the white and pial surfaces are compared separately. ↓ indicates smaller metric value is better, whereas ↑ indicates greater metric value is better.

| | | | T1w ✓ | ✗ | ✓ | ✓ | ✓ |
| | | | T2w ✗ | ✓ | ✓ | ✓ | ✓ |
| | | | RL ✓ | ✓ | ✓ | ✗ | ✓ |
| | | | Seg ✓ | ✓ | ✗ | ✓ | ✓ |
|---|---|---|---|---|---|---|---|
| CD ↓ | Pial | L | 0.73(±0.40) | 0.53(±0.09) | 0.38(±0.19) | 0.45(±0.22) | 0.35(±0.09) |
| | | R | 0.54(±0.12) | 0.68(±0.04) | 0.40(±0.12) | 0.56(±0.16) | 0.38(±0.13) |
| | White | L | 0.76(±0.28) | 0.57(±0.25) | 0.37(±0.14) | 0.51(±0.16) | 0.36(±0.14) |
| | | R | 0.54(±0.12) | 0.56(±0.23) | 0.37(±0.14) | 0.59(±0.18) | 0.38(±0.15) |
| ASSD ↓ | Pial | L | 0.63(±0.06) | 0.56(±0.08) | 0.49(±0.04) | 0.52(±0.04) | 0.47(±0.03) |
| | | R | 0.57(±0.05) | 0.63(±0.08) | 0.50(±0.05) | 0.57(±0.05) | 0.49(±0.05) |
| | White | L | 0.66(±0.05) | 0.58(±0.10) | 0.47(±0.05) | 0.55(±0.05) | 0.47(±0.05) |
| | | R | 0.58(±0.06) | 0.69(±0.09) | 0.48(±0.06) | 0.59(±0.05) | 0.49(±0.06) |
| HD ↓ | Pial | L | 1.23(±0.16) | 1.07(±0.16) | 0.87(±0.07) | 0.96(±0.10) | 0.81(±0.06) |
| | | R | 1.10(±0.13) | 1.23(±0.19) | 0.89(±0.09) | 1.11(±0.17) | 0.86(±0.09) |
| | White | L | 1.32(±0.12) | 1.15(±0.28) | 0.82(±0.09) | 1.06(±0.13) | 0.82(±0.09) |
| | | R | 1.14(±0.14) | 1.36(±0.20) | 0.85(±0.11) | 1.22(±0.14) | 0.87(±0.11) |
| NC ↑ | Pial | L | 0.85(±0.02) | 0.88(±0.02) | 0.91(±0.01) | 0.92(±0.01) | 0.92(±0.01) |
| | | R | 0.88(±0.02) | 0.85(±0.02) | 0.91(±0.01) | 0.89(±0.02) | 0.91(±0.01) |
| | White | L | 0.83(±0.03) | 0.91(±0.02) | 0.94(±0.01) | 0.92(±0.02) | 0.94(±0.01) |
| | | R | 0.89(±0.02) | 0.83(±0.02) | 0.94(±0.01) | 0.90(±0.02) | 0.94(±0.01) |

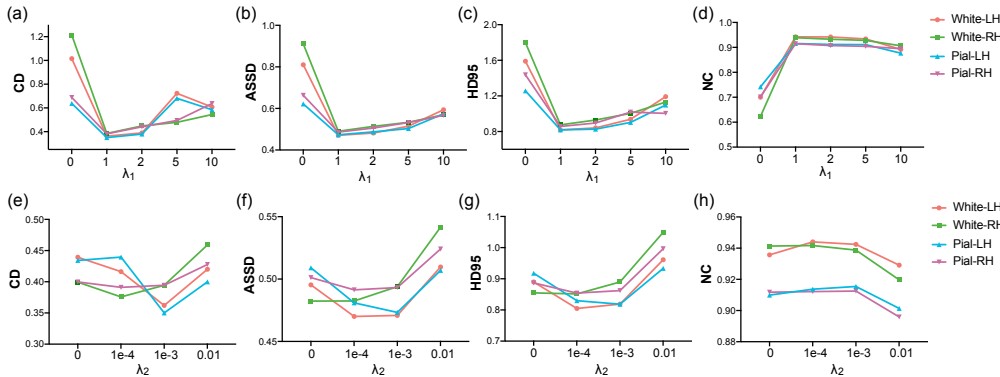

Figure 6: Impact of weights for AELL and segmentation loss on CSR performance.

quality of the generated surfaces, notably reducing irregular spikes in predictions when examined visually in the testing data.

We also conducted experiments to evaluate the impact of different values for $\lambda_1$ and $\lambda_2$. As shown in Fig. 6, it becomes apparent that selecting an appropriate value for $\lambda_1$ is crucial for achieving good performance. Values that are either too large or too small result in significantly larger CD, ASSD and HD values. In our experiments, we found that the optimal value for $\lambda_1$ is 1. Similarly, we determined that the best choice for $\lambda_2$ is 0.001.

## 5 CONCLUSION

In this paper, we presented a flow-based approach for efficient CSR and spherical mapping. To address the challenges arising from the substantial deformation required for deforming a spherical mesh template to the highly folded cortex, we propose a recurrent deformation learning strategy by gradually warping the mesh template to the white surface and subsequently to the pial surface. Experiments on the challenging task of CSR based on infant brain MRI demonstrate that our method significantly improves mesh regularity and reduces geometric errors compared to state-of-the-art deep learning approaches, while maintaining high time efficiency.

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
