# Supplementary Materials

## Detail of the network structure

Our method includes two sub-models with identical architecture for the reconstruction of the white and pial surfaces. We present the details of the models in Fig. 1 below.

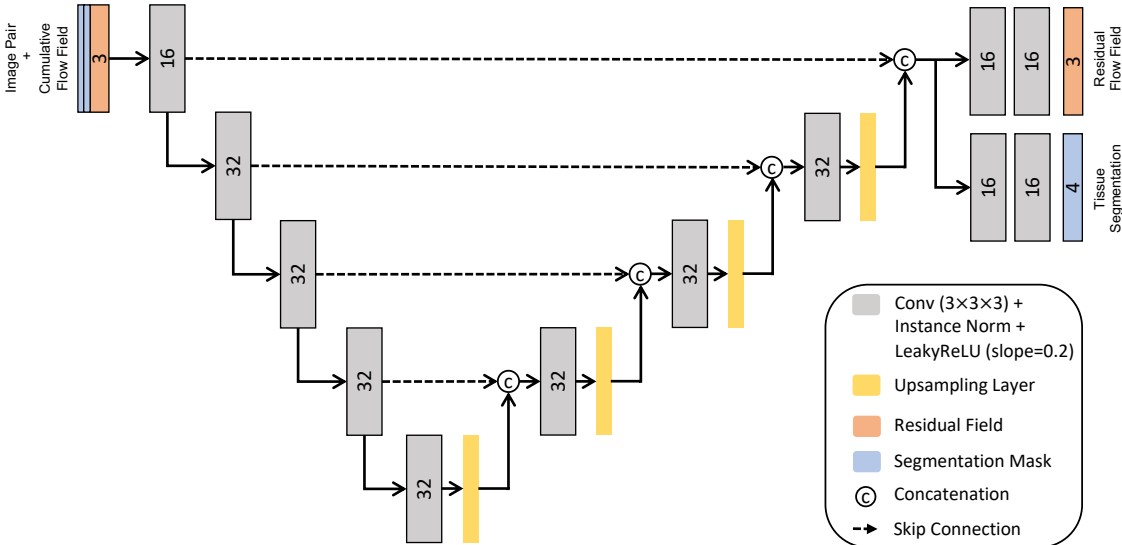

Figure 1: Common U-net architecture for the white and pial models.

## Segmentation Results

Dice coefficients were computed for gray matter, white matter, and cerebrospinal fluid with values of 0.883 ± 0.014, 0.885 ± 0.014 and 0.826 ± 0.028, respectively. Figure 2 shows segmentation results in comparison to ground truth (GT).

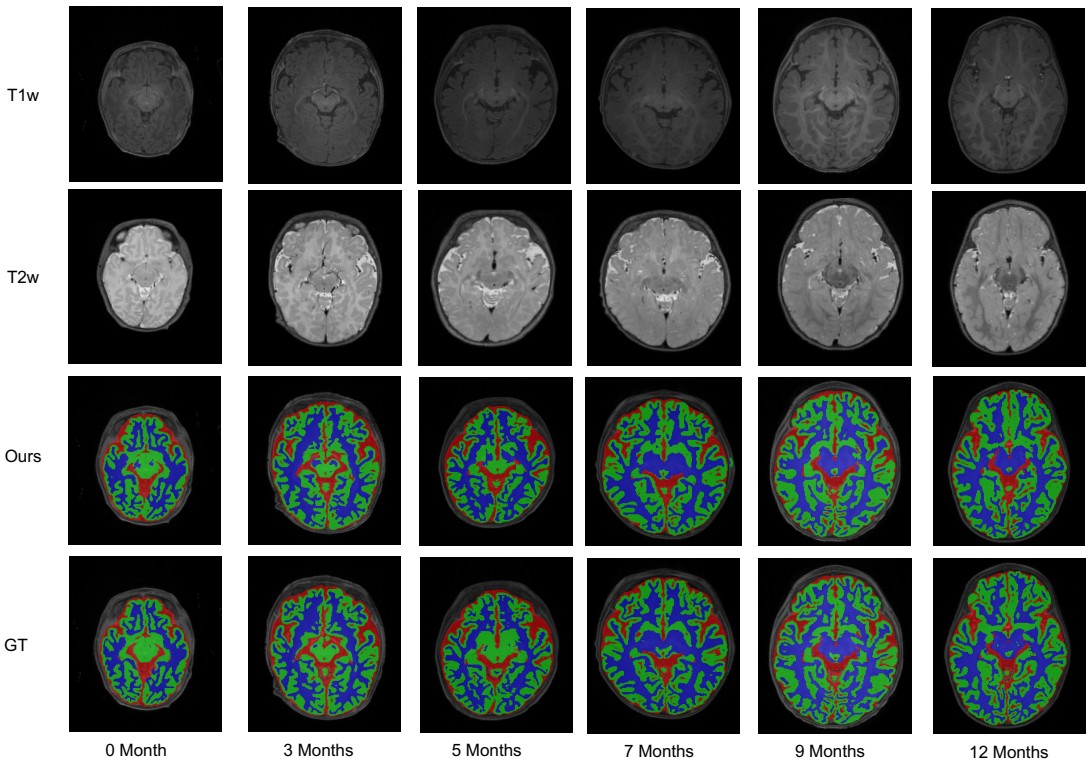

Figure 2: Segmentation results.