# OpenReview forum: "Reconstruction of Cortical Surfaces with Spherical Topology from Infant Brain MRI via Recurrent Deformation Learning"
_ICLR.cc/2024/Conference — Submitted to ICLR 2024_

### Official Review · Reviewer_o2xm · 2023-10-27

**Soundness:** 2 fair
**Presentation:** 2 fair
**Contribution:** 2 fair
**Rating:** 3
**Confidence:** 4

**Summary:**

Neuroanatomical studies of populations often perform analyses on the cortical surface of the brain (imaged by MRI), with the brain represented by a mesh (or a mesh-inflated-to-a-sphere) enabling the study of local cortical shape thickness. Submission 6817 presents a framework for the fast extraction of cortical surfaces using supervised learning to match the performance of a conventional solver.

In current literature, a popular approach is to train a supervised UNet-style network output multiple deformation fields that iteratively warp a template mesh into the desired ground truth cortical surface (with the ground truth obtained by FreeSurfer or similar). Submission 6817 follows this approach but instead uses a recurrent UNet instead of an unrolled UNet. Experimentally, it focuses on supervised infant cortical surface extraction using data from the Baby Connectome Project.

**Strengths:**

- The proposed “adaptive edge length loss” is an interesting and intuitive approach to dealing with the dramatic volume changes in the brain in the first year of growth.
- Using multi-modal T1w & T2w MRI as inputs for all considered methods is a nice touch as infant surface extraction has lower contrast and thus benefits from inter-modality differences.

**Weaknesses:**

### No methodological or implementation details provided:
As presented, this paper is unreproducible and hard to evaluate as no implementation and low-level modeling details are provided. For example, there’s no network architecture, no code, no details regarding dataset preprocessing (crucial to CSR), no details about data augmentation, how baselines were tuned, etc. The short implementation subsection lists learning rates, number of iterations, and loss weights, but spends the other half of its space describing instance normalization.

**Suggestion:** As the submitted paper doesn’t make it to the ICLR page limit (and there’s unbounded supplementary space), please actually describe the framework and its details and also share code if possible.

### ODE and iterative deformation contributions presented as novel:
Section 3.2 presents the proposed ODE-based deformation formulation which is framed as a contribution. However, this appears to be the same formulation as used in most previous work on CSR (e.g., CortexODE, CortexFlow, etc.) with the only distinction being that a recurrent UNet is used instead of an unrolled UNet.

**Suggestion:** Please detail how this specific contribution is non-trivially different from previous work and describe any specific innovations.

### Baseline performance does not match trends reported in previous work:
The experiments show the proposed method widely outperforming well-established baseline methods such as Vox2Cortex and CorticalFlow++. However, these experimental trends are highly unexpected as previous work also on infant data show that the differences between methods are in the order of 0.01 mm whereas this paper reports differences of 1mm or more (for example, see table 1 in CoTAN, MICCAI’23 https://arxiv.org/pdf/2307.11870.pdf).

How were these baselines implemented? Were their hyperparameters tuned on validation data ? If so, please add supplementary plots supporting this. Crucially, several of these methods start from an initial template cortical surface and not a sphere as in this method - were proper initial templates when these methods were run, or was a sphere used? If the latter, published hyperparameters for these methods will no longer transfer to the setting of this paper and need significant reworking for fair comparison.

**Suggestion:** Please clarify these experimental differences w.r.t. other published work and describe in detail how the baselines were implemented and tuned.

### Only a single dataset
The paper presents itself in the context of infant brain surface extraction which does not have many public datasets associated with it. However, none of the proposed methods in the submission are specific in any way to infants and could be directly applied to adults (in datasets such as HCP or ADNI) or neonates (e.g., the dHCP dataset). To actually evaluate the generalizability of the method, please include experiments on atleast one more dataset as is common in most papers in the subfield.

**Suggestion:** If experimenting on another dataset is not possible, please explain why the method is specific to infants in the 0-8 mo. age range.

### Overstretched claims
The paper makes some claims that are not well supported. For a few examples,
- The introduction (para. 3) claims that papers based on graph convolutions such as Hoopes, et al and Bongratz, et al. do not generate topologically valid surfaces and require topology correction. This is untrue as both methods perform the same topologically-valid approach as the submission (deforming a template to a target) and do not require correction.
- The paper claims that infant brain myelination (“tissue contrast changes”) poses a significant challenge to cortical surface reconstruction because of poor segmentation. However, this is only true in the unsupervised setting (e.g., using EM-style methods). As the paper trains its segmentor in a fully supervised manner, tissue segmentation is no more challenging than it is in adult brains as evidenced by all the work using the iSeg dataset.

**Suggestion:** Please temper these claims.

**Questions:**

Please see above, all of my suggestions to focus the rebuttal are highlighted in bold.

---

> ### Author Response · Authors · 2023-11-18
> **Responding the weaknesses pointed**
>
> **1. No methodological or implementation details provided**
>
> **Response:** We refer the reviewer to Figure S1 in the supplementary material for details on the common network architecture for the white and pial models. We added descriptions on data processing and more details on reproducing other baseline methods in Sections 4.1 and 4.2. Please let us know if more details are needed.
>
> **2. ODE and iterative deformation contributions presented as novel**
>
> **Response:** We want to address a potential misunderstanding regarding the contribution of our work. Our primary innovation lies in the integration of CSR and spherical mapping into a unified framework. Unlike existing CSR methods, our approach takes advantage of one-to-one vertex correspondence between the spherical template and cortical surfaces, eliminating the need for additional time-consuming spherical mapping and enhancing overall efficiency.
>
> While the ODE-based problem formulation and recurrent deformation learning strategy are not novel when considered individually, we want to highlight that we are the first to apply a recurrent learning strategy for CSR. This decision serves a dual purpose. Firstly, it optimizes parameter usage and GPU memory, resulting in a streamlined design that enables a single model to predict both white and pial surfaces. This contrasts with many existing methods, such as CorticalFlow++ and SurfFlow, which employ distinct models for each surface. Secondly, the recurrent learning strategy allows for the incremental learning of large deformations through multiple smaller deformations. This is crucial for our method, given the challenges posed by using a spherical template, where the initial template is farther from the target surfaces compared to segmentation-based initial surfaces or convex hulls, commonly used in existing deep learning-based methods.
>
> We apologize for any confusion caused by our initial description and have revised our paper to present our contributions more clearly.
>
> **3. Baseline performance does not match trends reported in previous work**
>
> **Response:** We attribute the observed challenges faced by baseline methods in our study to the unique characteristics of the dataset employed. The infant brain MRI images pose significant difficulties due to poor image contrast and dynamic variations in both contrast and brain volume. These challenges have not been adequately addressed by existing methods, as these methods are typically designed for adult brain MRI images where such issues are either absent or less pronounced.
>
> **4. A single dataset for evaluation**
>
> **Response:** Our primary goal is to solve an important but less-explored problem, which is to do CSR from infant brain MRI images. A vast majority of existing methods are developed for adult brain MRI images. Compared to adult brain MRI images, which exhibit good white matter vs. gray matter contrast, infant brain MRI images show much worse and dynamically changing image contrast and thus are significantly more challenging for CSR (see Figure S2 for some examples). In addition, the brain sizes are relatively stable for adult brains. However, the infant brain is fast growing. For example, the brain volume for a 12-month infant is approximately 2-3 times larger than a newborn (i.e., 0-month-old). Therefore, this issue is not well considered by existing methods.
>
> **5. Overstretched claims**
>
> **Response:**
>
>   1) We have adjusted the corresponding section in our paper to moderate this assertion, as recommended. While the authors of graph convolutional network-based works have noted that their methods do not require topology correction, it is important to clarify that these methods cannot ensure the topological correctness of the generated surfaces, even if the initial template surface is topologically correct. This is evidenced by our updated results in Table 1, which shows the self-intersection rates of the predicted surfaces by Vox2Cortex are significantly higher than flow-based methods, including ours.
>
>   2) In contrast to certain existing methods, such as CortexODE, which require the prior segmentation of the ribbon (i.e., subcortical-filled white matter), our approach directly reconstructs cortical surfaces from the image. Consequently, our model faces substantial challenges due to the poor image contrast.
>
>   3) We want to clarify that our method uses the segmentation as an auxiliary task mainly for additional regularization. During inference, the model does not require a segmentation map, thus the MR image quality will have a big impact on the results. In spite of this, our method showed decent performance even when the segmentation module is disabled, as we reported in the ablation study (see Table 2 in our paper).

---

> > ### Comment · Reviewer_o2xm · 2023-11-23
> > **Reviewer Response**
> >
> > Dear authors:
> >
> > Thank you for your response and revision. Unfortunately, due to the preliminary nature of the experiments and confusing claims, I will be maintaining my score. For some brief justification:
> >
> > #### Re:2 Novelty
> > IMO the submission does not clearly disambiguate what is its main new contributions vs. what is commonly done in or straightforward to do with prior work. For example, the rebuttal now clarifies that neither the recurrent nor the ODE formulation are its own contributions and that its main innovation is using a spherical template as input instead of a more customized brain template. This is a confusing claim as the considered baselines can just as well start from a spherical template to get the purported benefits of not needing post-processing. Other technical claims such as “Dual-Modal Input” (Sec 3.3) are also confusing as infant brains in the 0-12 mo. are always processed for CSR with both T1 and T2 images in existing literature so I do not understand the novelty.
> >
> > #### Re:3 Unexpectedly low baseline performance
> > The revision now states that “_All baselines are reproduced based on the recommended settings specified in their official implementation and paper_”. This is insufficient as all of those methods had their official hyperparameters tuned on adult brain MRI which will, in all likelihood, perform suboptimally on the infant cohort considered here. To support the claim of wide improvement on current work, the baseline hyperparameters need to be retuned on a validation split of this target dataset as there is a significant domain shift otherwise.
> >
> > #### Re:4 Only using a single dataset
> > I did not mention only adult brain MRI in the review, but also neonatal brains from dHCP that come with ground truth meshes for training. Neonatal brains go through much more dramatic size and shape changes than infants and could demonstrate the claimed utility of the method for large temporal changes. Further, the rebuttal claims that methods for adult brain CSR will not work for infant brain CSR. My review was talking about the opposite, i.e. evaluating the proposed method on other (potentially easier in the case of adult brains) datasets to evaluate the method in a generic setting as nothing about the proposed method is infant-specific methodologically.

---

> ### Author Response · Authors · 2023-11-23
> **Response to reviewer o2xm to clarify further**
>
> 1. It is thought by the reviewer that it is straightforward to use a spherical template in other baseline methods. However, this is not true for two reasons. Firstly, some methods impose stringent requirements on the initial surface. For instance, CortexODE requires the segmentation of the ribbon as a preliminary step, utilizing the segmentation mask to construct the initial surface. Altering this surface would fundamentally disrupt their pipelines, making it a nontrivial task. Secondly, opting for a spherical template amplifies the deformation required to align with the target surface. Current methods struggle with substantial deformations, hence their reliance on templates closer to the target. The recurrent deformation learning strategy we adopted is designed to tackle this issue, enabling our model to effectively handle such large deformations.
>
> 2. While we acknowledge there may still be room for improvement in the performance of competing methods, it's essential to underscore that we have exerted our utmost effort within the constraints of time and resources available to us.
>
> 3. As previously stated, our primary objective is to address a significant yet less-explored issue. Consequently, we directed our attention to CSR within what is arguably the most challenging age group—infants aged 0-12 months. It is noteworthy that, although neonatal brains undergo substantial changes in volume similar to infant brains, neonatal images from dHCP do not display the pronounced contrast changes observed in infant brains (refer to Figure 2 in the supplementary material).

---

### Official Review · Reviewer_o99L · 2023-10-28

**Soundness:** 3 good
**Presentation:** 3 good
**Contribution:** 2 fair
**Rating:** 3
**Confidence:** 4

**Summary:**

This paper proposes a recurrent method for cortical surface reconstruction by deforming a sphere template.

**Strengths:**

1. Utilize a general sphere template as the starting shape for the reconstruction task.
2. Utilize a recurrent method.
3. The presentation is clear and the experiments and visualization are sound.

**Weaknesses:**

1. This work is an incremental work based on the previous CSR methods, like CortexODE, Vox2Cortex, etc. Instead, this work utilized a more general sphere template, while other methods get the template from a mean shape or topology correction. Even though the author mentions that the sphere template is beneficial for spherical mapping, for the cortex reconstruction task, I don't see the advantage of using a sphere template for CSR.
2. The recurrent framework is not novel, as it has been applied to medical image registration for a long time. Even though it might be the first time introduced into the CSR, it is not novel enough for ICLR.
3. The proposed framework: template -> white surface -> pial surface, is pretty similar to CortexODE, with the change of using a sphere template and multiple-step surface reconstruction.

**Questions:**

The proposed method contains limited innovation and is mainly incremental work. Even though the author wants to emphasize the benefit of spherical mapping, from the CSR point of view, the method is not novel enough for ICLR.

**Details Of Ethics Concerns:**

No ethics concerns

---

> ### Author Response · Authors · 2023-11-18
> **Responding the weaknesses pointed**
>
> **1. Incremental work based on existing methods**
>
> **Response:** Our research builds upon established CSR methods, particularly those based on diffeomorphic flows. While we recognize the significant achievements of previous methods in reducing processing time from hours to mere seconds, our work seeks to contribute by addressing challenges that remain unresolved or insufficiently considered in existing approaches.
>
> Our primary contribution lies in the integration of CSR and spherical mapping. To achieve this goal, we introduce a recurrent deformation learning strategy to deform a spherical template onto the white and pial surfaces based on images. Unlike existing CSR methods, our approach leverages the one-to-one vertex correspondence between the spherical template and the cortical surfaces, requiring no extra time to achieve spherical mapping.
>
> Moreover, our method significantly enhances efficiency compared to all existing methods by eliminating the time-consuming spherical mapping step. However, it is essential to acknowledge that employing an initial template that is more distant from target surfaces, as opposed to using a convex hull or an initial surface based on cortical ribbon segmentation, significantly increases the level of difficulty. To tackle this challenge, we introduced a recurrent learning strategy to allow a large deformation to be learned incrementally with multiple smaller deformations. Notably, using ribbon segmentation to obtain an initial surface for deformation can lead to topological defects and thus requires topology correction, as no segmentation methods are perfect. In contrast, our method circumvents the error-prone segmentation by reconstructing the surfaces directly from images and requires no time-consuming topology correction as the flow-based approach can theoretically guarantee topological correctness.
>
> In summary, we believe our method signifies a significant stride forward in the field, serving as a pioneering solution in seamlessly integrating CSR and spherical mapping.
>
> **2. Benefits of the spherical template**
>
> **Response:** Many downstream tasks of CSR, such as cortical surface parcellation, require spherical mapping. For example, parcellation involves surface projection onto a sphere through metric-preserving inflation (i.e., spherical mapping) first and then registration to a spherical atlas. Notably, the inflation process alone takes about half an hour to complete and is only possible if a mesh has a homotopic topology to that of a sphere. The use of a spherical template allows our approach to take the advantage of one-to-one vertex correspondence between the spherical template and cortical surfaces. It not only guarantees that the topology of the predicted surfaces is homotopic to a sphere, but also eliminates the need for additional time-consuming spherical mapping and thus enhances overall efficiency significantly.
>
> In contrast, existing CSR methods often neglect spherical mapping in their algorithms, resulting in an inefficient pipeline. Our approach is distinct from the existing ones, addressing the inefficiencies of the traditional pipeline, and presenting a more streamlined and effective solution.
>
> For additional information on cortical analysis, we refer the reviewer to Figure 1 in our paper and related work below:
>
> [1] Hoopes, Andrew, et al. "TopoFit: rapid reconstruction of topologically-correct cortical surfaces." Proceedings of machine learning research 172 (2022): 508.
>
> [2] Gopinath, Karthik, Christian Desrosiers, and Herve Lombaert. "Learning joint surface reconstruction and segmentation, from brain images to cortical surface parcellation." Medical Image Analysis 90 (2023): 102974.
>
> **3. Recurrent learning not novel**
>
> **Response:** We want to emphasize again that we see the overarching concept of integrating CSR and spherical mapping as our main contribution, not the specific strategies, e.g., the recurrent deformation learning.
>
> Additionally, we want to mention that although the recurrent learning strategy is not new, we are the first to use it for CSR. Recurrent learning benefits both efficiency and effectiveness, allowing our method to handle both the white and pial surface reconstruction within a unified framework.
>
> **4. Multi-step strategy similar to existing works**
>
> **Response:** The multi-step strategy, progressing from the template to the white surface and culminating at the pial surface, has its roots in traditional CSR methods, such as FreeSurfer. This sequential approach is widely embraced across all CSR methods, except for those specifically tailored for exclusive white or pial surface reconstruction. Our method adopts this proven strategy in a deep learning framework.

---

### Official Review · Reviewer_Ciwf · 2023-10-30

**Soundness:** 2 fair
**Presentation:** 2 fair
**Contribution:** 2 fair
**Rating:** 5
**Confidence:** 4

**Summary:**

This paper proposed a recurrent deformation learning method for cortical surfaces reconstruction. The validation on the BCP dataset shows the good performance.

**Strengths:**

The results show improvement over the competing methods.

**Weaknesses:**

1.The details of the proposed mehthod are not clearly.
2. The topology correction is not validated.
3. The time consuming should be given.
4. The segmentation performance also should be given.

**Questions:**

1.The details of the proposed mehthod are not clearly.
2. The topology correction is not validated.
3. The time consuming should be given.
4. The segmentation performance also should be given.

---

> ### Author Response · Authors · 2023-11-18
> **Responding the weaknesses pointed**
>
> **1. Details on the proposed method are not clear**
>
> **Response:** We refer the reviewer to Figure S1 in the supplementary material for details on the common network architecture of the white and pial models. Please let us know if more details are needed.
>
> **2. Topology correctness is not validated**
>
> **Response:** To evaluate topology correctness, we reported the self-intersection (SI) rates for all methods in the table below. Our method demonstrates a relatively low SI rate (~0.3% on pial surfaces and ~0.06% on white surfaces). Notably, DeepCSR and CortexODE outperform in this metric. DeepCSR benefits from using the marching cubes algorithm for final predictions rather than deforming a template, so it is expected to have a low SI rate. CortexODE achieves a lower SI rate possibly due to the use of initial surfaces closer to target surfaces than the spherical template employed in our method. In comparison, all other methods exhibit significantly poorer performance than ours.
>
> For reference, we calculated the SI rate on our pseudo-ground truth, revealing that the pseudo-ground truth (GT) meshes are far from perfect. It also suggests that the performance of our proposed method in terms of the SI rate may be underestimated, potentially influenced by the pseudo-ground truth.
> Additionally, we computed the Euler characteristic of our method's predicted surfaces, all of which equal to 2, indicating correct genus-zero topology in our predictions.
>
> SI (%) for pial surface:
> |   | Ours         | DeepCSR      | CorticalFlow++ | SurfFlow  | Vox2Cortex | CortexODE |   GT      |
> | - | ------------ | ------------ | -------------- | --------- | ---------- | --------- | --------- |
> | L | 0.30土0.14    | 0.001土0.0005 | 1.30土0.42      | 1.43土0.52 | 8.15土2.22  | 0.23土0.11 | 4.12土0.93 |
> | R | 0.38土0.15    | 0.002土0.0008 | 1.21土0.34      | 2.63土0.57 | 6.47土1.60  | 0.21土0.11 | 3.91土1.12 |
>
> SI (%) for white surface:
> |   | Ours         | DeepCSR      | CorticalFlow++ | SurfFlow  | Vox2Cortex | CortexODE |   GT      |
> | - | ------------ | ------------ | -------------- | --------- | ---------- | --------- | --------- |
> | L     | 0.06土0.06     | 0.001土0.0005 | 0.11土0.06 | 0.38土0.13      | 11.24土5.70 | 0.04土0.04   | 0.13土0.38 |
> | R     | 0.05土0.03     | 0.001土0.0006 | 0.12土0.07 | 1.43土0.43      | 10.83土5.58 | 0.003土0.003 | 0.02土0.02 |
>
> **3. The time consumed should be given**
>
> **Response:** The inference time of our method is 1.2 seconds for predicting the white and pial surfaces in the same hemisphere. It is noteworthy that our method saves a substantial amount of time for downstream tasks because no extra time is needed for spherical mapping. CorticalFlow++ and SurfFlow inference time are at the same level (~1 seconds) for CSR. However, they typically require an extra 30 minutes for spherical mapping.
>
> The CortexODE inference process involves multiple steps, including ribbon segmentation, topology correction, SDF computation, initial surface reconstruction via marching cubes, and surface deformation. The total computing time for all key steps is 5.5 seconds, covering the prediction of white and pial surfaces in the same hemisphere.
>
> The Vox2Cortex inference time is 17 seconds, encompassing all four surfaces in both hemispheres.
>
> **4. The segmentation performance also should be given**
>
> **Response:** We reported the segmentation performance and a visual comparison with ground truth of Figure 2 in the supplementary material. The average dice coefficient values of graymatter, white matter, and cerebrospinal fluid are 0.883土0.014, 0.885土0.014 and 0.826土0.028, respectively.

---

### Author Response · Authors · 2023-11-18

We sincerely thank the reviewers for their thoughtful and thorough review of our work. The insights and comments provided are invaluable, and we are grateful for the time and effort reviewers invested in offering constructive feedback. The incorporation of the suggestions has significantly enhanced the overall quality of our work.

We hope that our response has addressed reviewers’ questions. If there are any additional points that require clarification, please do not hesitate to let us know.

---

### Meta-Review · Area_Chair_WDJT · 2023-11-28

**Metareview:**

This paper proposed a recurrent deformation learning method for cortical surfaces reconstruction. Experimental results show that the proposed method outperforms the competing methods.

However, as the reviewers reported, the paper lacks technical details and the description of the methods is also quite rough. Besides, the framework of the proposed method is incremental, the main novelty may be lie in the using of sphere template, which should be not enough for ICLR. Moreover, there are concerns about the about the fairness of the experiment, which may cause the baseline performance does not match trends reported in previous work.

**Justification For Why Not Higher Score:**

I agree with the reviewers’ comments that the novelty of this paper is insufficient for ICLR.

**Justification For Why Not Lower Score:**

The recommendation is to reject the paper.

---

### Decision · Program_Chairs · 2024-01-16

Reject